# Retinoic Acid Grafted to Hyaluronic Acid Activates Retinoid Gene Expression and Removes Cholesterol from Cellular Membranes

**DOI:** 10.3390/biom12020200

**Published:** 2022-01-25

**Authors:** Vojtěch Pavlík, Veronika Machalová, Martin Čepa, Romana Šínová, Barbora Šafránková, Jaromír Kulhánek, Tomáš Drmota, Lukáš Kubala, Gloria Huerta-Ángeles, Vladimír Velebný, Kristina Nešporová

**Affiliations:** 1R&D Department, Contipro, a.s., 562 04 Dolní Dobrouč, Czech Republic; veronika.machalova@contipro.com (V.M.); martin.cepa@contipro.com (M.Č.); romana.sinova@contipro.com (R.Š.); barbora.safrankova@gmail.com (B.Š.); jaromir.kulhanek@contipro.com (J.K.); tomas.drmota@centrum.cz (T.D.); Gloria.Huerta-Angeles@contipro.com (G.H.-Á.); vladimir.velebny@contipro.com (V.V.); kristina.nesporova@contipro.com (K.N.); 2Third Faculty of Medicine, Charles University, 100 00 Prague, Czech Republic; 3Institute of Experimental Biology, Faculty of Science, Masaryk University, 625 00 Brno, Czech Republic; kubalal@ibp.cz; 4International Clinical Research Center, St. Anne’s University Hospital, 656 91 Brno, Czech Republic

**Keywords:** retinoic acid, hyaluronic acid, amphiphilic hyaluronan, nanocarrier, cholesterol, HyRetin, Delcore

## Abstract

All-trans-retinoic acid (atRA) is a potent ligand that regulates gene expression and is used to treat several skin disorders. Hyaluronic acid (HA) was previously conjugated with atRA (HA-atRA) to obtain a novel amphiphilic compound. HA-atRA forms micelles that incorporate hydrophobic molecules and facilitate their transport through the skin. The aim of this study was to determine the influence of HA-atRA on gene expression in skin cells and to compare it with that of unbound atRA. Gene expression was investigated using microarrays and a luciferase system with a canonical atRA promoter. HA-atRA upregulated gene expression similarly to atRA. However, HA-atRA activated the expression of cholesterol metabolism genes, unlike atRA. Further investigation using HPLC and filipin III staining suggested that the treated cells induced cholesterol synthesis to replenish the cholesterol removed from the cells by HA-atRA. HA modified with oleate (HA-C18:1) removed cholesterol from the cells similarly to HA-atRA, suggesting that the cholesterol removal stemmed from the amphiphilic nature of the two derivatives. HA-atRA induces retinoid signaling. Thus, HA-atRA could be used to treat skin diseases, such as acne and psoriasis, where the combined action of atRA signaling and anti-inflammatory cholesterol removal may be potentially beneficial.

## 1. Introduction

All-trans-retinoic acid is an essential signaling molecule in embryogenesis, cellular growth, differentiation, and apoptosis [1]. atRA and its precursors (such as retinal and retinol) are ingested as vitamin A. The precursors can be metabolized to atRA, which then signals through its nuclear receptors (RARs, PPARβ/δ) that dimerize and regulate gene expression in association with co-activators or co-repressors. The receptor dimers with atRA bind to DNA motifs called retinoic acid response elements (RARE). Apart from the direct influence of gene expression, atRA exerts nongenomic effects, such as the activation of kinase networks, that integrate with genomic effects [2]. Clinicians use atRA (together with other retinoid derivatives) to treat several types of cancer and skin diseases [3,4,5]. Topically applied atRA on UV-exposed skin alleviated the signs of photoaging and increased collagen production and epidermal thickness [6]. On the other hand, atRA is banned in cosmetics in the EU because of the side effects that may occur. In contrast, less potent retinoids with fewer adverse events, such as retinol, retinyl aldehyde, and retinyl palmitate, are allowed in topical formulations [7]. Safe atRA derivatives with high potency are constantly sought-after.

HA is a linear, non-sulfated glycosaminoglycan composed of altering disaccharides of D-glucuronic acid and N-acetyl-D-glucosamine. Cartilages, vitreous, and skin contain considerable amounts of HA. The molecular size of HA in the extracellular matrix of skin is mostly considered as 4-6 MDa (HMW-HA) [8]. HMW-HA has several structural roles. It creates a backbone for protein attachment, and its high water-retention capacity maintains skin viscoelasticity and lubricates joints and fasciae [9]. Oppositely, low-molecular-weight HA (LMW-HA) elicits different cellular responses, such as fibrosis, photoaging or wound healing [10]. LMW-HA, unlike HMW-HA, penetrates the *stratum corneum* [11]. HA is a subject of crosslinking by TNF alpha-induced protein 6 (also known as TSG-6), resulting in complexes that regulate cellular responses differently than a linear HA [12]. 

HA can be conjugated with fatty acids, such as oleyl or lauroyl, yielding biocompatible derivatives with amphiphilic properties [13,14]. Such derivatives form micelles that incorporate hydrophobic compounds into their core and facilitate transport through the skin. In a previous work, the conjugation of atRA and HA was reported (HA-atRA) in a procedure that allowed for precise control of the degree of substitution [15,16]. HA-atRA was found to be safe for fibroblasts (1000 µg/mL, 48 h), whereas the corresponding dose of unbound atRA was found to be highly cytotoxic [15]. Micelle-forming HA-atRA proved to be an efficient nanocarrier through the epidermis. However, the effects elicited by HA-atRA in human skin cells remain unknown. In addition, since atRA potently upregulates gene expression of many targets, the differences in transcriptional responses to HA-atRA and atRA need to be addressed. Such information is helpful for the evaluation of the potential therapeutic uses of HA-atRA.

We therefore evaluated whether HA-atRA can induce gene-expression changes comparably to atRA using targeted luciferase assay and broad-spectrum transcriptome profiling with microarrays. The results underscore the similar and unique properties of HA-atRA compared to atRA.

## 2. Materials and Methods

### 2.1. Cell Treatment

HA (13 kDa) and its derivatives were manufactured in Contipro (Dolní Dobrouč, Czech Republic). HA-atRA was prepared using 13, 79 or 270 kDa HA, with the degree of substitution (DS) ranging between 5.5 and 6.3%. The derivative was characterized by NMR, UV-VIS and SEC-MALLS, as described elsewhere [15] and in the Appendix A. HA-atRA was stored at −20 °C, and the stability of the derivative was checked regularly. HA grafted with oleic acid (HA-C18:1) with DS 13% (also known under the brand name Delcore, manufactured in Contipro) was used in the tests. 25-hydroxycholesterol (≥98%) was obtained from Cayman Chemical (Ann Harbor, MI, USA), methyl-β-cyclodextrin (MβCD) was purchased from Santa Cruz Biotechnology (Santa Cruz, CA, USA), TGFβ1 was obtained from Promokine (Heidelberg, Germany), and ethanol was obtained from Penta (Prague, Czech Republic). All-trans-retinoic acid (98%), retinyl palmitate, retinyl acetate (≥90%), U18666A (≥95%) and DMEM (Dulbecco’s modified Eagles medium, low glucose) were purchased from Merck Life Science UK Ltd. (Dorset, UK).

HA-atRA was dissolved in DMEM cell-cultivation medium [supplemented with glutamine (0.3 mg/mL), glucose (4 mg/mL), penicillin (100 units/mL) with streptomycin (0.1 mg/mL) and 10% fetal bovine serum (FBS)] for 1 h at 37 °C, protected from light. The solution was subsequently filtered through a 0.22 µm filter. The final concentration of HA-atRA and HAC18:1, unless otherwise specified, was 100 µg/mL and 50 µg/mL, respectively. HA-C18:1 was dissolved overnight at room temperature in a cell-cultivation medium with 10% serum. Ethanol was used to dissolve atRA in concentrations corresponding to the molarity of atRA bound to HA-atRA; 4.86 µg/mL atRA was used in the experiments in which 100 µg/mL HA-atRA was applied. 25-hydroxycholesterol was dissolved in ethanol to 10 mM, and the stock was stored at −20 °C for a maximum of 2 months. Methyl-β-cyclodextrin was dissolved in a serum-free cultivation medium. Cells were rinsed twice with PBS before the 15 min treatment with MβCD (final concentration: 2.5 mM). TGFβ1 was dissolved to 30 ng/mL in sterile water containing 0.1% BSA and stored in aliquots at −80 °C. Retinyl palmitate and retinyl acetate were dissolved in 96% ethanol.

### 2.2. Cell Cultivation

Normal human dermal fibroblasts from adult skin (NHDF) were purchased from Lonza (Basel, Switzerland). Fibroblasts were cultivated in a medium supplemented with 10% FBS in 75 cm^2^ culture flasks under 5% CO_2_ and at 37 °C until the fifth passage. The HaCaT keratinocyte cell line was obtained from Hölzel Diagnostika (Köln, Germany) and was cultivated, like NHDF, without the addition of glucose to the medium. Cells were sub-cultured two to three times per week after reaching 80% confluency using 0.25% trypsin.

### 2.3. Luciferase Assay

P19 cells that stably expressed an atRA-sensitive luciferase reporter in a pRAREβ2-TK-luc plasmid were cultured as previously described [17,18]. Cells were cultured on tissue culture dishes pre-treated for 5 min using a 0.1% aqueous solution of gelatin from porcine skin (Merck Life Science UK Ltd., Dorset, UK) in Dulbecco’s modified Eagle’s medium containing 10% fetal calf serum, 0.05 mM β-mercaptoethanol, penicillin and streptomycin.

P19 cells were seeded at 5 × 10^5^ into a 6-well plate pre-treated with gelatin. After 24 h, the cells were treated with the specified compounds corresponding to the molarity of free atRA. The cells were rinsed with PBS and lysed (100 µL of lysis buffer, 15 min incubation at RT) 6 h after the treatment. The cell lysates were scraped with a rubber policeman and transferred into a new tube. The luciferase activity was determined according to the manual for the luciferase assay (Luciferase Reporter Gene Assay, high sensitivity, Roche, Basel, Switzerland). The luciferase activity was measured immediately (sample by sample) by measuring 20 µL of a sample mixed with 100 µL of luciferase substrate. The signal was acquired by a PerkinElmer (Waltham, MA, USA) plate reader as luminescence intensity integrated over 3 s. Protein concentration was measured using a BCA protein assay kit (ThermoFisher Scientific, Waltham, MA, USA) in doublets of 10 µL aliquots of the cell lysate. The luminescence signal was normalized to protein concentration and related to the untreated control.

### 2.4. RNA Isolation

HaCaT cells (1.7 × 10^5^) were seeded in 6 wells and left untreated for 48 h. NHDF cells were seeded at a concentration of 2 × 10^5^/6-well and treated after 24 h. The cultivation medium was aspirated, the cells were lysed using 350 µL of RLT buffer contained in an RNeasy Mini Kit (Qiagen, Düsseldorf, Germany), and total RNA was isolated according to the manufacturer’s instructions. RNA concentration and quality were assessed spectrophotometrically (NanoDrop One, ThermoFisher Scientific, Waltham, MA, USA).

### 2.5. Microarray

NHDF cells (1.6 × 10^6^) were seeded into 75 cm^2^ flasks and left to adhere overnight before treatment. Then, NHDF cells were treated with HA-atRA (corresponding to 16.14 µM atRA, which is equal to 100 µg/mL of HA-atRA DS 5.8%) or atRA for 24 h. Control samples were left untreated. RNA was isolated as described above using 700 µL RLT lysis buffer. RNA quantity, purity and quality were assessed spectrophotometrically and via Agilent 2100 Bioanalyzer capillary electrophoresis. A total of 500 µg RNA was labeled with Cy3 or Cy5 (GE Healthcare, Chicago, IL, USA) and amplified according to the instructions in the manual for the Low Input Quick Amp Labeling Kit (Agilent, Santa Clara, CA, USA). Briefly, 900 µg of the labeled aRNA was hybridized using a total-RNA gene-expression chip (Human Gene Expression v2 4x44K Microarray Kit, G4845A, Agilent, Santa Clara, CA, USA) for 17 h. The slides were processed as described in the manual and scanned with a GenePix 4000B scanner (Molecular Devices, San Jose, CA, USA).

Data were processed in R studio [19]. The limma software package was used to analyze the gene-expression values [20]. Quantile normalization was applied. *p* values were adjusted to multiple testing with the Benjamini-Hochberg method, and the mean log2 fold changes (FC) of the treated samples vs. untreated controls were calculated. Gene-set ontology of the genes with *p* < 0.05 and log2 FC > 1 was analyzed using g:Profiler [21].

### 2.6. qPCR

cDNA synthesis was performed using the High-Capacity cDNA Reverse Transcription Kit (ThermoFisher Scientific, Waltham, MA, USA) with 1 µg of total RNA as a template in 20 µL reactions. The resulting cDNA samples were diluted at 1:100 before adding to qPCR reactions. Real-time PCR was performed using TaqMan Fast Advanced MasterMix (ThermoFisher Scientific, Waltham, MA, USA) under fast cycling conditions using StepOnePlus (ThermoFisher Scientific, Waltham, MA, USA) real-time PCR cycler. The following TaqMan assays (ThermoFisher Scientific, Waltham, MA, USA) were used: RPL13A (Hs04194366_g1), HMGCS1 (Hs00940429_g1), SQLE (Hs01123768_m1), DHRS3 (Hs01044021_m1), SREBF2 (Hs01081784_m1) and LDLR (Hs01092524_m1).

Data were normalized to RPL13A1 mRNA levels, and gene-expression fold change was calculated using the 2^∆∆Ct^ method. The statistical significance of the differences between the treated and control samples (each *n* = 5) was compared using Student’s *t*-test (Excel, Microsoft Office).

### 2.7. siRNA Treatment

HaCaT cells were seeded by a similar process to that used as in the gene-expression experiments. siRNA against SREBF2 (s29, ThermoFisher Scientific, Waltham, MA, USA) was dissolved to a 5 µM concentration and stored at −20 °C. For each well in a 6-well plate, 8.8 µL SREBF2 siRNA or control siRNA (AllStars Negative Control siRNA, SI03650325, Qiagen, Düsseldorf, Germany) was mixed with 200 µL jetPrime buffer (PolyPlus, Illkricht, France) and vortexed. Then, 8 µL of jetPrime reagent was added, vortexed and incubated for 10 min at RT. Subsequently, the mixture was added dropwise to the culture medium (with serum) and slightly shaken to ensure even distribution. After 72 h, 1 mL of HA-atRA (100 µg/mL) dissolved in culture medium was added to the culture medium with siRNA for 24 h, so the siRNA was present for the whole experiment. Gene-expression analysis was performed in the same way as described above.

### 2.8. Western Blotting

HaCaT cells were treated in 6-well panels for 24 h. The proteins were harvested, and the protein concentrations were measured using the BCA protein assay kit. Western blots against PCNA (clone PC10, sc-56, Santa Cruz Biotechnology, Santa Cruz, CA, USA) and HMGCS1 (clone D5W8F, #36877, Cell Signaling Technology, Danvers, MT, USA) were performed. Both primary antibodies were used in 1:1000 dilution (overnight, 4 °C), and secondary antibody was diluted to 1:2000 (1 h, RT; 7076S (anti-mouse), 7074S (anti-rabbit), Cell Signaling Technology, Danvers, MT, USA). The membranes were developed with HRP-conjugated anti-rabbit or anti-mouse Ig using chemiluminescent detection to visualize signals (Clarity Western ECL substrate, Bio-Rad, Hercules, CA, USA). The signals were scanned and evaluated with the Alliance 9.7 Chroma Chemiluminescence Imaging System (UVItec Limited, Cambridge, UK). The signal for each derivative-treated sample was related to the untreated control. The statistical significance of the differences between the treated and control samples (each *n* = 5) was compared using Student’s *t*-test (Excel, Microsoft Office).

### 2.9. Cholesterol Quantification with HPLC

HaCaT cells were seeded onto large flasks (75 cm^2^) and were either treated with 10 mL of media with HA derivatives or left untreated. After cultivation, the cell medium was aspirated for cholesterol quantification. The flasks were rinsed twice with PBS. Then, 1.2 mL of a lysis buffer (50 mM Tris, 150 mM NaCl, 0.1% SDS, 1% Triton X-100, pH 8.0) was added, and the cells were scraped. The samples were sonicated to reduce viscosity. Part of the sample (200 µL) was used to measure protein concentration. Then, each remaining 1 mL of sample was mixed with 4 mL of 25% KOH in ethanol in HadeSpace vials. After vigorous shaking, the vials were incubated at 85 °C for 60 min. The vials were mixed thoroughly twice during the incubation. Afterward, the samples were left to cool at room temperature. Then, 1.3 mL water and 4 mL heptane were added. The mixture was vortexed for 40 s. The two phases separated after 10 min at RT. The upper (heptane) phase with cholesterol was further analyzed for cholesterol content. Calibration solutions were prepared by dissolution of cholesterol in heptane (500 µg/mL), followed by dilution, to obtain calibration points (5–100 µg/mL). The samples, calibration solutions and blanks were analyzed using an SFC Waters HPLC system with a PDA detector.

The samples were separated on an ACQUITY UPC2 Torus 1-AA, 130Å, 1.7 µ (3.0 mm × 100 mm) column preheated to 40 °C. The samples were measured at 205 nm. The samples were separated with supercritical CO_2_, methanol with mobile phase flow (2 mL/min) and a methanol gradient (2–10%). Analysis time was 4.5 min. One microliter of each sample was injected onto the column.

Chromatograms were processed using Empower 3 software. The resulting cholesterol concentrations were normalized to protein concentration in the samples. The significance of the change in the cholesterol was assessed compared to the untreated control using two-sample Student’s *t*-test (Excel, Microsoft Office).

### 2.10. Filipin III Staining and Quantification

HaCaT keratinocytes were seeded (7 × 10^4^ cells/chamber of a glass 8-chamber histological slide) and left to adhere for 6 h, then treated with U18666A in the specified wells for 48 h. Subsequently, the cells were treated with HA-atRA and HA-C18:1 for another 24 h. After that, the cells were fixed with 1.5% paraformaldehyde applied for 10 min, washed three times with PBS and incubated with glycine dissolved in PBS (c = 1.5 mg/mL) for another 10 min. Next, the cells were washed once with PBS and incubated with filipin III, which binds to membrane cholesterol, using the Cholesterol Assay Kit, Cell-Based (Abcam, Cambridge, UK) according to the manual. The cells were then washed three times with PBS.

The labeled cells were imaged using a Leica TCS SP8 X confocal microscope (Wetzlar, Germany) equipped with a Leica HC PL APO CS2 objective (63×, 1.20 NA, water) at 405 nm laser line for filipin III excitation. Brightfield images were acquired with the same illumination, using transmitted light detection. The images were acquired as z-stacks (2 µm step). All images were processed using FiJi software. Maximum intensity projection (MIP) was performed in the fluorescence channel, and foreground/background was segmented in the brightfield channel. Segmented brightfield images were then overlaid over fluorescence MIP images, selecting only areas in the images that contained cells. The mean fluorescence intensity of filipin III in such areas was estimated. Images from at least three positions were acquired in each well.

## 3. Results

### 3.1. HA-atRA Has a Similar Ability to Induce Gene Expression as atRA

The effect of atRA conjugated with HA (HA-atRA) on gene expression was evaluated using P19 cells (a mouse pluripotent cell line). The cell line stably expresses a luciferase reporter activated by atRA through a pRAREβ promoter. HA-atRA was evaluated alongside the unbound atRA and other commonly used and topically applied retinoids (retinyl palmitate and retinyl acetate). Since HA-atRA has amphiphilic properties, we included an HA ester with oleic acid, HA-C18:1, in the experiments.

HA-atRA retained its ability to induce gene expression initiated via the pRAREβ2 promoter in a dose-dependent manner (Figure 1). A higher dose of HA-atRA (5 µM atRA equivalents) was needed to achieve the effect elicited by 1 µM atRA. However, even higher doses of retinyl acetate (RA) were needed to achieve the effects of 1 µM HA-atRA. RP induced only negligible luciferase expression compared to the other retinoids. HA-C18:1 and HA served as controls, which, as expected, did not induce luciferase expression; their luminescence values corresponded to the untreated cells.

We further investigated whether HA-atRA changes gene expression differently than atRA using gene expression microarrays. Human-skin fibroblasts were treated, and their transcription profiles were compared to the untreated control. HA-atRA upregulated (>200% expression of the untreated control, *p* < 0.05) gene expression of 43 genes, and atRA upregulated 87 genes under the same criteria. Almost all genes upregulated by atRA were also elevated in HA-atRA (Figure 2, Appendix A). Among the genes upregulated in both groups were typical genes known to be elevated by atRA, such as DHRS3 (dehydrogenase/reductase 3), which catalyzes the reduction of atRA to retinol [22], and RARRES1 (retinoic acid receptor responder 1) or LXN (latexin), both of which participate in the anticancer activity of atRA [23].

### 3.2. HA-atRA Upregulates Expression of Cholesterol Synthesis Genes

However, in the microarray experiment, 18 genes were significantly upregulated only in the HA-atRA group (Figure 2, Appendix A). Twelve of the genes were classified using the gene-set ontology analysis belonging to the cholesterol biosynthetic process (Appendix A), and even the remaining six genes have functions in cholesterol homeostasis.

qPCR was used to measure the gene expression of a subset of HA-atRA-upregulated genes involved in the biosynthesis of terpenoid and steroid: HMGCS1 (hydroxymethylglutaryl-CoA synthase), SQLE (squalene epoxidase) and FDPS (farnesyldiphosphate synthase). Gene expression was analyzed in both major skin-cell types: fibroblasts and keratinocytes. The cells were treated with atRA, HA-atRA and HA-C18:1 to investigate whether the upregulation of steroid synthesis genes is caused by a unique atRA moiety bound to HA or whether it stems from the amphiphilic properties of this molecule. HA-atRA upregulated the expression of HMGCS1, SQLE and FDPS to the same extent as HA-C18:1, whereas atRA did not induce such elevation (Figure 3). Additionally, HA-atRA and HA-C18:1 significantly elevated protein levels of HMGCS1 (Appendix A). These results suggest that the effect induced by HA-atRA occurs independently of the biological properties of atRA. A more likely scenario is that the physicochemical properties of the HA ester grafted with hydrophobic molecules were responsible. The gene-upregulation effect was not restricted to HA-atRA derived from 13 kDa HA but was also observed using HA-atRA synthesized with 79 or 270 kDa HA (Appendix A). DHRS3 was upregulated by HA-atRA and not by HA-C18:1, confirming that HA-atRA maintains the ability to induce atRA-mediated canonical gene expression, also seen in the luminescence reporter experiment.

The upstream regulation of HMGCS1 and other cholesterol metabolism genes is governed by the SREBP2 transcription factor (sterol regulatory element binding transcription factor 2, encoded by SREBF2 gene) [24]. SREBF2 expression, together with its downstream genes, HMGCS1 and LDLR (low-density lipoprotein receptor), was induced by HA-atRA and HA-C18:1 (Figure 4). We therefore employed an inhibitor of SREBP2 activation, 25-hydroxycholesterol (25HC) [25], to investigate the direct involvement of SREBP2 in the upregulation of cholesterol metabolism induced by the HA derivatives. The inactive form of SREBP2 is restrained in the endoplasmic reticulum due to 25HC binding, and therefore, SREBP2 is not processed into its active form in the Golgi apparatus. Treatment with 25-HC led to the downregulation of SREBF2, HMGCS1 and LDLR transcription, even in the presence of HA-atRA or HA-C18:1 (Figure 4). Moreover, the targeting of SREBF2 by siRNA decreased its expression to 23% relative to the untreated control. The cells treated with HA-atRA and siRNA had lower expression of HMGCS1 than those treated with HA-atRA only (Appendix A). These results show that cells treated with HA-atRA or HA-C18:1 exploit SREBP2 for the upregulation of cholesterol genes.

### 3.3. Cells Treated with HA-atRA or HA-C18:1 Contained Less Cholesterol

As the SREBF2 transcription factor and the downstream genes of cholesterol synthesis were upregulated, we quantified cellular cholesterol after treatment with HA-atRA or HA-C18:1 using HPLC (Figure 5A). Surprisingly, cellular cholesterol levels were significantly lower after the HA-atRA or HA-C18:1 treatment than in the untreated control. On the contrary, cholesterol content in the cultivation media was significantly higher after the treatments with the HA derivatives. Total cholesterol contents (cell lysates + the corresponding cultivation media) were higher in the HA-atRA- or HA-C18:1-treated samples, corresponding to the increased gene expression of cholesterol metabolism genes. 

Relative cholesterol content was quantified using filipin III staining via confocal microscopy (Figure 5B,C). The treatments with HA-atRA or HA-C18:1 resulted in significantly lower staining intensity, in concordance with the HPLC results. When cells are preincubated with the U18666A lysosomal cholesterol export inhibitor, the intracellular cholesterol increases [26]. Therefore, the loss of intracellular cholesterol in the U18666A-preincubated cells was even more pronounced after treatment with HA-atRA or HA-C18:1.

The content of cellular cholesterol is tightly regulated. Excessive cholesterol is actively transported from cells by ABCA1 and ABCG2 [27]. While cholesterol synthesis genes SREBF2 and HMGCS1 were upregulated, ABCA1 expression was significantly downregulated by more than 90% after treatment with HA-atRA or HA-C18:1 (Figure 6). ABCG2 expression did not significantly change in HA-C18:1 and HA-atRA samples (data not shown). This suggests that the cholesterol produced by the cells was transferred to the cultivation medium by a mechanism different from that acting with ABCA1-mediated efflux. In addition, the decrease in ABCA1 may be interpreted as cells minimizing the cholesterol loss by the canonical efflux pathway. As a control, we used TGFβ, which is known to induce cholesterol efflux via ABCA1 upregulation. TGFβ upregulated ABCA1, SREBF2 and HMGCS1, reflecting the situation when cholesterol efflux is increased and cells replenish its levels by increased synthesis.

Cellular cholesterol may have been removed from plasmatic membranes through HA-atRA or HA-C18:1, which could have induced cholesterol synthesis with the aim of replenishing the lost cholesterol. Methyl-β-cyclodextrin (MβCD) is used to acutely remove cellular-membrane cholesterol. Cells that were treated with MβCD and subsequently left to recover exhibited increased gene expression of SREBF2 and HMGCS1, although to a lesser extent than cells treated with HA derivatives or TGFβ (Figure 6).

## 4. Discussion

Retinyl palmitate and retinyl acetate are considered safe for use in cosmetics up to certain concentrations, whereas use of atRA is restricted. However, the efficacy of RP and RA is limited, as they need to be metabolized to atRA [28]. Our results confirm the lower activation of pRAREβ-mediated gene expression by RA, whereas RP did not induce gene expression. On the other hand, HA-atRA induced gene expression in the luciferase assay at concentrations comparable to those induced by atRA. The reduced activity of RA and RP may also be caused by differing bioavailability of these retinoids; since RP is very hydrophobic, RA exerts less hydrophobicity, whereas HA-atRA is amphiphilic and soluble in polar solvents. Bjerke et al. observed the activation of RARα reporter by retinol or retinyl propionate, whereas RP did not activate retinoid signaling in the HEK293 reporter cell line [29]. In addition, since RP is a storage form of atRA, levels of which are tightly regulated, only some cell types have retinyl ester hydrolases that can efficiently mobilize retinoids from RP [30].

HA-atRA upregulated genes known to be elevated by atRA, such as DHRS3, RARRES1 and LXN (Appendix A). In addition, the activation of signaling typical of atRA was retained at a similar magnitude in gene expression microarrays with regards to HA-atRA. However, the upregulation of steroid synthesis genes by HA-atRA relative to the untreated control was unexpected, as cholesterol synthesis was reported to be downregulated after atRA treatment in keratinocytes [31]. Our microarray data also show the downregulation of cholesterol synthesis genes as a result of atRA treatment (Figure 2, Appendix A). Further investigations suggested that HA-atRA and HA-C18:1 promoted the removal of cellular cholesterol to the cultivation media. In light of these findings, the upregulation of cholesterol synthesis and absorption from medium was found to reflect diminished cholesterol levels.

Since cholesterol is excreted from cells via ABCA1, we stimulated ABCA1 upregulation using TGFβ, resulting in decreased cellular cholesterol [32]. While TGFβ significantly upregulated ABCA1, the cells also upregulated SREBF2, HMGCS1 and LDLR. The upregulation of cholesterol synthesis as a result of TGFβ treatment was described previously [33]. In contrast to TGFβ, HA-atRA and HA-C18:1 significantly decreased the gene expression of ABCA1, suggesting other mechanisms of cholesterol removal.

Cholesterol may be removed from the plasmatic membrane by MβCD, which results in increased cholesterol synthesis and the inhibition of ABCA1-mediated efflux [34]. Zhao et al. also showed that treating MDA-MB-231 cancer cells with 2-hydroxypropyl-β-cyclodextrin leads to the upregulation of cholesterol synthesis and LDLR, as well as reduced ABCA1 expression [35]. Therefore, cyclodextrins exert similar effects as those observed with HA-atRA or HA-C18:1. However, while MβCD upregulated cholesterol synthesis genes via SREBP2 in our experiments, we observed no downregulation of ABCA1 expression. In our experiments, MβCD was applied for 15 min due to its cytotoxicity, whereas HA derivatives and TGFβ were applied for 24 h.

The exact mechanism of cholesterol removal mediated by the HA derivatives is unclear. HA-atRA and HA-C18:1 may have acted as the acceptors of free cholesterol or may have facilitated cholesterol exchange between cells and lipoproteins, similarly to MβCD [36]. Both HA-atRA and HA-C18:1 can form micelles that carry hydrophobic compounds. Micelles are established when concentrations of the derivatives are above the critical aggregation constant (CAC) [37]. Since the derivatives were present in cell-cultivation media at concentrations greater than the CAC, micelles may have formed. In addition, HA-C18:1 has been shown to interact with cellular membranes and decrease their fluidity [38].

MβCD is typically used to treat cells for between 30 min and 8 h, depending on cell types and MβCD concentrations, leading to 60% cholesterol depletion [39]. The HA derivatives removed 33% (HA-atRA) and 27% (HA-C18:1) cholesterol after 24 h. However, MβCD has cytotoxic side effects, and safer cyclodextrins are sought after [40]. We did not observe a loss of viability of the cells treated with the HA derivatives. Therefore, HA-atRA/HA-C18:1 could be a safer alternative or a complement to MβCD treatment. A closer comparison of cholesterol-removal dynamics needs to be conducted in future research. We focused on the synthesis of cholesterol, which may not be the only lipid affected by treatment with HA derivatives. Similarly, MβCD removes other lipids besides cholesterol [41].

Cholesterol accumulation in situ has been described to have a role in several human pathologies, such as atherosclerosis and Niemann—Pick Type C-1 disease (NPC). Brown et al. observed cholesterol efflux from NPC-patient cells mediated by the micelles of distearoyl-phosphatidylethanolamine-PEG [42]. Using filipin III staining, we showed that HA-atRA and HA-C18:1 removed cholesterol from the membranes of cells pre-treated with U18666A. This compound is used to mimic the cholesterol-efflux dysfunction of NPC disease [26]. Therefore, HA-atRA or HA-C18:1, per se or in conjunction with other drugs, present biodegradable carriers that can be considered for the treatment of NPC.

Macrophages in atherosclerotic plaques are defective in cholesterol efflux, rendering them pro-inflammatory [43]. Cholesterol depletion by a nanocarrier comprising MβCD diminished atherosclerotic plaques in vivo [44]. Cholesterol removal via HA-atRA or HA-C18:1 could also lower inflammation in atherosclerotic plaques. In addition, the anti-inflammatory effect could be beneficial in chronic wounds or other skin inflammatory diseases.

The stability and pharmacokinetics of the HA derivatives affect their systemic behavior and, therefore, their potential use. Although we have not investigated the stability of HA-atRA in vivo, Šimek et al. recently showed that HA-C18:1 administered intravenously in mice was stable in the bloodstream. The derivative was cleared from the body within 72 h, mainly through the activity of the liver [45]. Since the ester bond conjugating atRA to HA is similar to that conjugating C18:1 to HA, HA-atRA would likely have similar pharmacokinetics in vivo. Nevertheless, the proposed anti-inflammatory effects and cholesterol-removal abilities need to be investigated using suitable in vivo models.

## 5. Conclusions

We found that HA-atRA activates canonic gene expression similarly to atRA. Therefore, HA-atRA and atRA share the therapeutic potential to treat skin diseases and conditions such as acne, post-inflammatory hyperpigmentation and photoaging. However, the advantages and disadvantages over other retinoids need to be investigated in future research. HA-atRA and HA-C18:1 possess a novel and unique (compared to atRA) property, which lies in the removal of cellular cholesterol. Therefore, HA derivatives may be helpful in the mitigation of hypercholesteremia or atherosclerotic plaques.

## Figures and Tables

**Figure 1 biomolecules-12-00200-f001:**
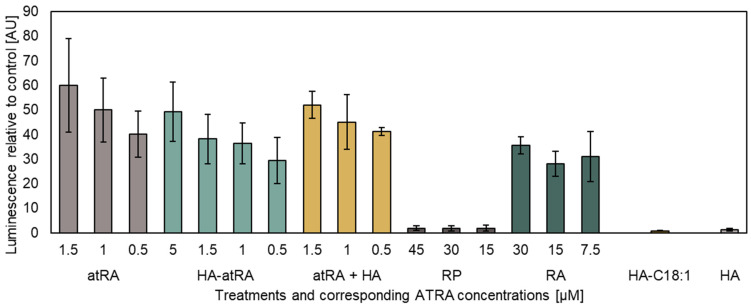
The luciferase reporter assay of atRA and its derivatives. Luminescence was related to untreated control P19 cells stably expressing the pRAREβ2 luciferase vector. The concentrations of the tested compounds containing atRA correspond to the micromoles of atRA present in the media. The concentrations of HA-C18:1 and HA were adjusted according to HA-atRA. HA-atRA: an ester of HA and retinoic acid; atRA: retinoic acid; atRA + HA: a physical mixture of HA and retinoic acid; RP: retinyl palmitate; RA: retinyl acetate; HA-C18:1: ester of HA and oleic acid; HA: hyaluronic acid. The bars represent the mean ± SD of *n* = 3 independent experiments. Luminescence was adjusted to protein total cell mass in each sample.

**Figure 2 biomolecules-12-00200-f002:**
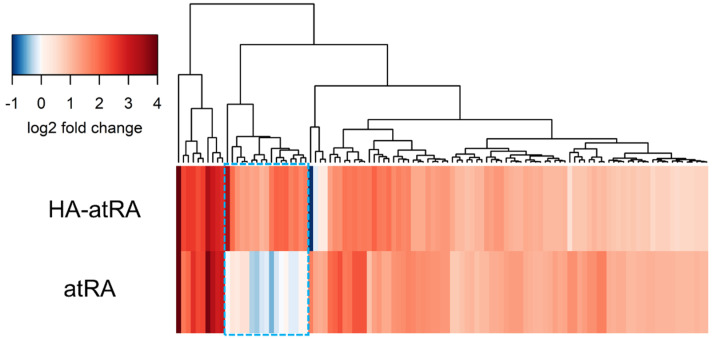
Genes upregulated by HA-atRA and atRA relative to control. Fibroblasts were treated with HA-atRA or atRA. Gene expression was analyzed using microarrays. The dendrogram shows the genes upregulated more than 200% in either or both groups relative to untreated control. Log2 of 1 corresponds to 200% upregulation; log2 of 2 is 400%, etc. Negative values of log2 represent downregulation. Box with blue dashed line highlights the genes significantly upregulated by HA-atRA only. N = 4 for each sample.

**Figure 3 biomolecules-12-00200-f003:**
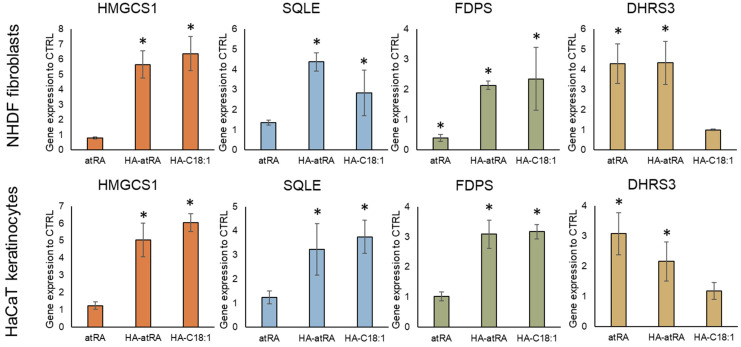
Gene expression of steroid synthesis genes (HMGCS1, SQLE, FDPS) and canonical atRA-induced gene DHRS3. Keratinocytes and fibroblasts were treated with atRA, HA-atRA and HA-C18:1. Gene expression was measured by means of qPCR, and the values are related to untreated controls. The bars represent the mean ± SD of four independent experiments. * *p* < 0.05, *t*-test compared to untreated control.

**Figure 4 biomolecules-12-00200-f004:**
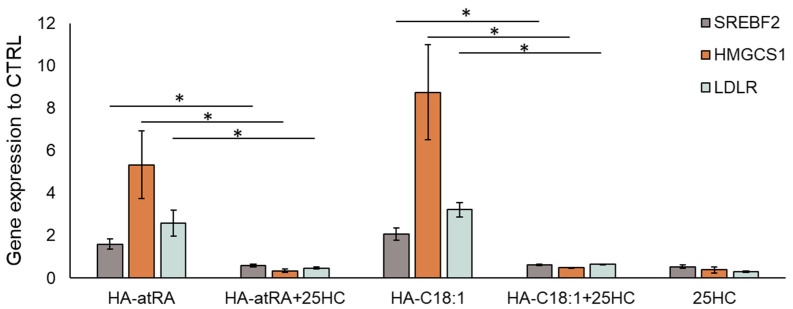
Inhibition of SREBP2 by 25-HC in HaCaT keratinocytes. Gene-expression analysis of the samples treated with HA-atRA or HA-C18:1 combined with 25-hydroxycholesterol (1 µM). The bars represent the mean ± SD of four independent experiments. The decreases in SREBF2, HMGCS1 and LDLR after the addition of 25-HC were significant (* *p* < 0.05, *t*-test) compared to respective HA-atRA and HA-C18:1 treatments.

**Figure 5 biomolecules-12-00200-f005:**
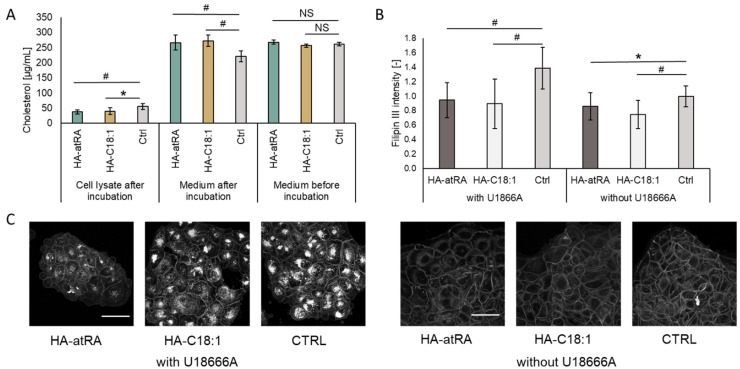
Cholesterol content in HaCaT cells incubated with HA-atRA of HA-C18:1 for 24 h. (**A**) Cholesterol was measured in cell lysates and corresponding media before and after incubation with HA derivatives using HPLC. Bars represent the mean ± SD of n = 6 experiments. (**B**) HaCaT cells were pre-incubated for 48 h with or without U18666A inhibitor and treated for 24 h with HA-atRA or HA-C18:1 or left untreated; cholesterol was stained with filipin III, and the signal was quantified. Bars represent the mean ± SD of n = 6 experiments. (**C**) A representative set of images of cells stained with filipin III. All six images are of the same magnitude. The scale bars correspond to 50 µm. * *p* < 0.05, # *p* < 0.01; NS, not significant, *t*-test.

**Figure 6 biomolecules-12-00200-f006:**
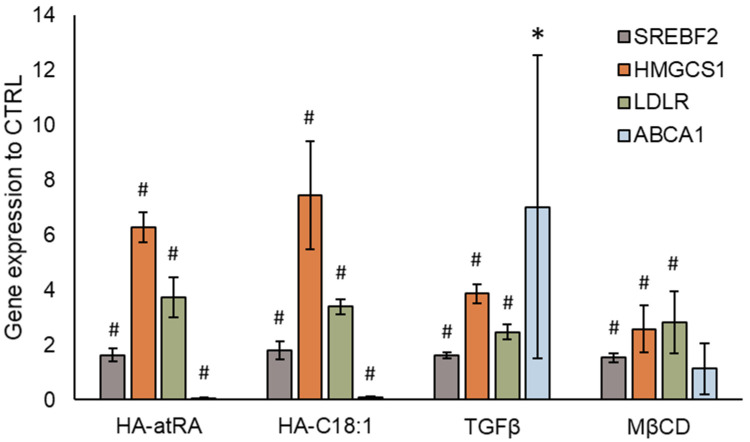
Gene expression of cholesterol transporter ABCA1. Cholesterol synthesis was induced by HA-atRA, HA-C18:1, TGFβ or MβCD. The bars show the mean ± SD of *n* = 4 experiments relative to the untreated controls. * *p* < 0.05, # *p* < 0.01 compared to the untreated control using *t*-test.

## Data Availability

Not applicable.

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
