# Peer review of "Retinoic Acid Grafted to Hyaluronic Acid Activates Retinoid Gene Expression and Removes Cholesterol from Cellular Membranes"

_biomolecules, 2022, doi:10.3390/biom12020200_

Round 1
Reviewer 1 Report
The manuscript "Retinoic acid grafted to hyaluronan activates retinoid gene expression and removes cholesterol from cellular membranes" written by Pavlik V, Machalova V, Čepa M, Šinova R, Šafrankova B, Kulhanek J, Drmota T, Kubala L, Huerta-Angeles G, Velebny V and Nešporova K, describes the effects of all-trans-retinoic acid conjugated with hyaluronic acid on cultures of keratinocytes and fibroblasts. Activation of reporter gene regulated with retinoic acid inducible promoter showed HA-ATRA compound to have similar activity as ATRA. Microarray analysis showed also upregulation in expression of a set of genes regulated by ATRA, but, in addition, changes in expression of a set of genes involved in cholesterol metabolism. Furthermore, experiments showed that hyaluronic acid compounds removed cholesterol from the cells.
The manuscript has interesting topic. It is well written, analyses correctly designed and presented appropriately. Materials and methods are detailed, Results clearly presented and Discussion is adequate.
Minor comments:
Lines 12, 53, 72, 87: full names of HA (hyaluronic acid), TSG-6, DS, HAC18:1
Results could have subtitles
Line 322: Figure 4 instead of 1
Line 363: Methyl-beta.cyclodextrin was used...
Line 382: "since RP is a stable form of ATRA..." better explanation
Line 440: Full stop missing.
Author Response
Thank you for the time you devoted to improve our manuscript. We are pleased to hear that the message was clearly communicated. We address your comments and suggestions below.
Minor comments:
Lines 12, 53, 72, 87: full names of HA (hyaluronic acid), TSG-6, DS, HAC18:1
We added the definition of abbreviation HA at its first use (abstract). Also, we changed "hyaluronan" in the title to hyaluronic acid to correspond better with further use. TSG-6 abbreviation was amended as it is now recognized as TNF Alpha Induced Protein 6. The degree of substitution was defined at its first use. HA-C18:1 abbreviation was defined also in methods.
Results could have subtitles
The subtitles were added.
Line 322: Figure 4 instead of 1
The pdf file generated by the submission system and distributed to reviewers contains this error. The original manuscript in word is correct. We will prudently control the proof version.
Line 382: "since RP is a stable form of ATRA..." better explanation
Stable was substituted with storage.
Line 440: Full stop missing.
The full stop was added.
Reviewer 2 Report
Use "Hyaluronic Acid (HA)" instead Hyaluronan in the title and text, and "atRA" instead ATRA.
The Authors investigated the effect on gene expression in skin cells of a Hyaluronic Acid (HA) conjugated Retinoic Acid, a multifunctional vitamin (A) essential in the human body for embryogenesis to adulthood. Multiple phenotypic expressions effects of vitamin A are exerted mainly by the all-trans retinoic acid (atRA), an metabolite regulating the expression of target genes by nuclear receptors such as (RARs, RXRs, PPARβ/δ ...), and other coregulators. It should be mentioned that atRA exerts nongenomic effects such as the activation of kinase networks that integrate with genomic effects (extranuclear and non-transcriptional effects integrated in the nucleus via the phosphorylation). The paper compares gene expressions human skin fibroblast from HA-atRA and unbound atRA. The up-regulation of cholesterol synthesis and absorption is one of the points that needs more attention and referencing of existing literature.
These aspects should be clearly stated in the discussion and conclusions.
Author Response
Thank you for the time you devoted to the review of our manuscript and for the helpful comments that improve the manuscript.
Use "Hyaluronic Acid (HA)" instead Hyaluronan in the title and text, and "atRA" instead ATRA.
The changes were made accordingly.
The Authors investigated the effect on gene expression in skin cells of a Hyaluronic Acid (HA) conjugated Retinoic Acid, a multifunctional vitamin (A) essential in the human body for embryogenesis to adulthood. Multiple phenotypic expressions effects of vitamin A are exerted mainly by the all-trans retinoic acid (atRA), an metabolite regulating the expression of target genes by nuclear receptors such as (RARs, RXRs, PPARβ/δ ...), and other coregulators.
It should be mentioned that atRA exerts nongenomic effects such as the activation of kinase networks that integrate with genomic effects (extranuclear and non-transcriptional effects integrated in the nucleus via the phosphorylation).
This is already addressed on lines L35-36.
The paper compares gene expressions human skin fibroblast from HA-atRA and unbound atRA. The up-regulation of cholesterol synthesis and absorption is one of the points that needs more attention and referencing of existing literature. These aspects should be clearly stated in the discussion and conclusions.
The upregulation of cholesterol synthesis exerted by cyclodextrins was expanded in the discussion and referenced accordingly (L407-410).
Reviewer 3 Report
This study is interesting and the manuscript is well written; however, minor comments listed below should be addressed as follows:
- Significant findings should be highlighted in the abstract. Line 22: it should be (HA-ATRA induces retinoid signaling; it could thus be used………………..).
- The novelty of the study should be emphasized at the end of the introduction before mentioning the aim of the study.
- Have the authors performed statistical analyses of the obtained data? Please add a section for statistical analyses, demonstrating the employed software and adopted tests at the end of the methodology. Additionally, how many replicates have been done over the entire experiments?
- Line 322: It should be corrected to be (Figure 4). Please add scale bars to Figure 5c.
- Some other related work might be cited and discussed to improve this work; for instance, (https://doi.org/10.1016/j.msec.2018.04.053; https://doi.org/10.3390/ma11040569;https://doi.org/10.1016/j.jbiotec.2020.02.00).
Author Response
Thank you for your insightful comments and spotted mistakes. We address them below.
This study is interesting and the manuscript is well written; however, minor comments listed below should be addressed as follows:
- Significant findings should be highlighted in the abstract. Line 22: it should be (HA-ATRA induces retinoid signaling; it could thus be used………………..).
The expression in the abstract was amended.
- The novelty of the study should be emphasized at the end of the introduction before mentioning the aim of the study.
We added the information about the gap in current research that should be addressed and how the information contained in the presented manuscript contributes to the issue. Then we stated the aim of the study.
- Have the authors performed statistical analyses of the obtained data? Please add a section for statistical analyses, demonstrating the employed software and adopted tests at the end of the methodology. Additionally, how many replicates have been done over the entire experiments?
Statistical analyses were performed for gene expression microarrays (details are given in the methods sections including the correction for multiple testing), we apologize for the missing information on the number of replicates in the microarray experiment, it was added at the end of Figure 2 caption. The treatment of data resulting from gene expression measurements by means of qPCR as well as the statistical evaluation is described in methods and the captions of Figures 3,4, and 6. The changes in cholesterol levels using HPLC were evaluated using t test in six replicates as specified in the caption of Figure 5 and methods (Cholesterol quantification with HPLC). Statistical analysis was not performed in the luminiscence experiment as this was more phenomenological experiment, where the data showed a decrease depending on the dilution. The luminiscence experiment was performed thrice as indicated in the figure caption.
We feel that the statistical information given in the figure captions and descriptions accompanying the abovementioned methods serves its purpose better than a separate chapter in methods.
- Line 322: It should be corrected to be (Figure 4). Please add scale bars to Figure 5c.
The pdf file generated by the submission system and distributed to reviewers contains this error. The original manuscript in word is correct. We will prudently control the proof version.
Scale bars were added.
- Some other related work might be cited and discussed to improve this work; for instance, (https://doi.org/10.1016/j.msec.2018.04.053; https://doi.org/10.3390/ma11040569; https://doi.org/10.1016/j.jbiotec.2020.02.002).
Thank you for the suggestions. We expanded the discussion on cholesterol replenishment after its removal from cells with other references closer to the topic.